# Quantitative Multiplex Real-Time Reverse Transcriptase–Polymerase Chain Reaction with Fluorescent Probe Detection of Killer Immunoglobulin-Like Receptors, *KIR2DL4/3DL3*

**DOI:** 10.3390/diagnostics10080588

**Published:** 2020-08-13

**Authors:** Wipaporn Wongfieng, Rungtiwa Nutalai, Amonrat Jumnainsong, Chanvit Leelayuwat

**Affiliations:** 1Research Administration Division of Khon Kaen University, Khon Kaen 40002, Thailand; wongfieng@hotmail.com; 2The Centre for Research and Development of Medical Diagnostic Laboratories (CMDL), Faculty of Associated Medical Sciences, Khon Kaen University, Khon Kaen 40002, Thailand; rnutalai@gmail.com (R.N.); amonrat@kku.ac.th (A.J.); 3Biomedical Sciences Program, Graduates School of Khon Kaen University, Khon Kaen 40002, Thailand; 4Department of Clinical Immunology and Transfusion Sciences, Faculty of Associated Medical Sciences, Khon Kaen University, Khon Kaen 40002, Thailand

**Keywords:** fluorescent dyes, killer-cell immunoglobulin-like receptors, limit of detection, qRT-PCR

## Abstract

(1) Background: *KIR2DL4/KIR3DL3* are the framework genes present in all KIR haplotypes, with unique expression patterns being present only in women and CD56bright NK cells. *KIR* genes have a high degree of DNA sequence identity. Consequently, they are one of the most challenging genes for molecular detection—especially regarding expressions; (2) Methods: We developed an effective method to determine *KIR3DL3/KIR2DL4* expressions based on a multiplex quantitative real-time Reverse transcription polymerase chain reaction (qRT-PCR )with fluorescent probes using NK92; (3) Results: Standardizations of the singleplex *KIR2DL4* and *KIR3DL3* were performed to evaluate the sensitivity and specificity for further development of the multiplex assay. The limit of detection was at 500 copies each. There was cross-amplification with the presence of related *KIR* genes at a level of 5 × 10^7^ copies. This is not biologically significant because this high level of KIR expression has not been found in clinical samples. The multiplex assay was reproducible equivalent to its singleplex (*KIR2DL4*; *R*^2^ = 0.995, *KIR3DL3*; *R*^2^ = 0.996, but lower sensitivity of 10^3^ copies). Furthermore, the validation of the developed method on samples of blood donors showed high sensitivity (100%) and specificity (99.9%); (4) Conclusions: The developed method is reliable and highly specific suitable for evaluation of the *KIR2DL4/3DL3* mRNA expressions in further applications.

## 1. Introduction

Natural killer cells defend at the first line of innate immunity against tumorigenic and viral infected cells. NK cells are capable of killing the major histocompatibility complex (MHC)-Class-I-deficient target cells with defective engagement of inhibitory NK cell receptors [1,2]. There are three distinct families of the inhibitory NK receptors (iNKRs) in humans: the killer-cell immunoglobulin-like receptors (KIR), the leukocyte Ig-like receptors (LILR) [3] and the lectin-like CD94: natural killer group 2 (NKG2) family [4]. KIRs (KIR—also known as CD158) are major MHC Class I receptors that contain two (KIR2D) or three (KIR3D) Ig domains called D0, D1 and D2. The cytoplasmic domain can be subdivided into either inhibitory with immunoreceptor tyrosine-based inhibitory motifs (ITIM) (KIR2DL and KIR3DL) or activating receptors (KIR2DS and KIR3DS) which have a positively charged residue within a shorter cytoplasmic tail [5,6]. The KIR-mediated regulations can significantly elicit immune responses toward cancer, viral infection, hematopoietic stem cell transplantation and pregnancy [7,8,9].

The *KIR* genes are tandemly located within the leukocyte receptor complex on the human chromosome 19q13.4. These genes are highly complex for three reasons: (I) allelic variations; (II) the diverse level of surface expression and (III) haplotypic diversity and gene copy number variation. Previous analyses showed that there are A and B haplotypes of the *KIR* gene organization. More than 50 distinct *KIR* haplotypes have been identified [10,11,12]. The genomic organization of *KIR* haplotypes contains four framework genes—*KIR3DL3*, *3DP1*, *2DL4* and *3DL2*—which are present on all human *KIR* haplotypes. However, the stochastic KIR expressions generate clonalities of individual’s NK cells [13]. Unlike other *KIRs*, the two framework *KIR* genes have a unique expression pattern. First, *KIR2DL4* (CD158d) contains only one ITIM in the long cytoplasmic domain and a charged residue-linked adaptor molecule of FcεRI in its transmembrane region allowing this receptor to display both inhibitory and activating signals for NK cell responses. This structure is dominantly unique among the KIR family members. However, the mechanism that controls *KIR2DL4* to elicit activating or inhibitory signals yet remains unclear. During early stage of pregnancy, NK cells that are a majority of lymphocytes comprising in the fetal-maternal interface, play an important role for successful implantation. Indeed, *KIR2DL4* is one of the dominant receptors in uterine and decidual NK cells modulating pregnancy circumstances [14]. Its ligand, HLA-G, is expressed by the fetal trophoblast [15], by tumor cells [16] and by thymic epithelial cells [17]. Interestingly, the transcriptional level of *KIR2DL4* significantly decreases with increased risk for pre-eclampsia [13,14]. Moreover, the *KIR2DL4* surface expression on the uterine NK cells is expressed at a statistically higher level in a recurrent spontaneous abortion than normal controls. Second, *KIR3DL3* shows structural protein divergence away from *other KIR3DL*, with a deletion of the stem region and possesses one ITIM in its truncated cytoplasmic tail (as does *KIR2DL4*) [5,18]. An unusual feature of *KIR3DL3* is that the mRNA level is restricted and only expressed in the decidual NK and CD56^bright^ NK cells. However, the protein expression is absent. Moreover, women’s PBMCs tend to highly express the mRNA levels compared to those of men [19]. The role of *KIR3DL3* in pregnancy involving immune responsive NK cells is not yet clear. The expression pattern of *KIR3DL3* is not well examined because mRNA transcript is very difficult to detect in the total RNA from human peripheral blood mononuclear cells (PBMCs). In addition, the high degree of sequence similarities among *KIRs* makes them difficult to specifically be detected. To elucidate the function of these two *KIR* genes, it is essential to specifically detect their patterns of expressions in various cell types. The purpose of this study was to set up the quantitative real-time reverse transcription polymerase chain reaction (qRT-PCR) with fluorescent probe assay to carry out the *KIR2DL4/3DL3* mRNA detections. Novel probes specific for *KIR2DL4/3DL3* mRNAs have been developed. The novel method could simultaneously and comparably determine the expressions of *KIR2DL4/3DL3* in a single tube of a sample using a new reference gene, *RPII*. The method has been verified for sensitivity, specificity, limit of detection and cross-reactivity as well as validated on human samples. The developed multiplex real-time RT-PCR with the fluorescent probe assay could reliably differentiate biologic mRNA expression levels of *KIR2DL4* and *3DL3* in a single tube.

## 2. Materials and Methods

### 2.1. Sequence Analyses and Probe Design

All alleles of 17 *KIR* gene sequences were retrieved from the IPD-KIR Database [20]. Nucleotide sequences were multiple aligned and analyzed by Clustal Omega [21] which only one sequence was used to be a representative for all alleles of each KIR; *2DL1*001* (L41267.1), *2DL2*0010101*(U24075.1), *2DL3*0010101*(L41268.1), *2DL4*00101*(X99480.1), *2DL5A*0010101*(AF204903.1), *2DS1*001*(X89892.1), *2DS2*0010101*(L41347.1), *2DS3*00,101*(L76670.1), *2DS4*0010101*(U24077.1), *2DS5*001*(L76672.1), *3DL1*0010101*(U30274.1), *3DL2*0010101*(L41270.1), *3DL3*00,101*(AJ938052.1), *3DS1*010*(L76661.1). The numbers of alleles for each *KIR* gene were represented in Appendix A. All 109 alleles of *KIR2DL4* and 166 alleles of *KIR3DL3* were analyzed by BioEdit to design highly specific probes that could not cross-react to all alleles of other KIRs. All primers for *KIR* gene amplifications were used as described by Tajik el al., 2009 [22]. Whereas RNA polymerase II (*RPII*) sequences were retrieved from the NCBI Database and were used to design primers and probe to amplify the mRNA transcript at the exon 26–27, as a reference gene for normalization. Primer and probe sequences used in this study were given in Table 1.

### 2.2. Reverse Transcription, Polymerase Chain Reaction and Plasmid Constructs

The total RNA of NK92 was prepared by the TRIzol extraction (Invitrogen, Carlsbad, CA, USA) and consecutively conversed to cDNA by using the Moloney murine leukemia virus reverse transcriptase (M-MLV RT) (Promega, Madison, WI, USA) according to the manufacturer’s instructions. To generate plasmids of *KIR2DL4*, *KIR3DL1*, *KIR3DL3* and *RPII*, amplifications of segmented cDNA with specific primers (Table 1) were performed by the PCR assay. Each 25-µL reaction contained 500 ng of cDNA of NK92, 0.2 mM dNTP, 1X PCR buffer, 2 mM MgCl_2_, 0.3 µM of primers and 5 units of *Taq* DNA polymerase. The amplification cycles were; denaturation for 5 min at 95 °C, then 30 cycles of 30 s at 95 °C, 30 s at 60 °C and 30 s at 72 °C followed by 72 °C for 10 min. The PCR products purified by the illustra™ gel band purification kit (GE Healthcare, Chicago, IL, USA) were cloned into the pGem^®^T easy vector system (Promega, Madison, WI, USA) according to the manufacturer’s instructions and then transformed into the *E. coli* XL-1 blue cells (Stratagene, La Jolla, CA, USA). Positive clones were selected by ampicillin resistant-positive and blue-white colony selection and confirmed by DNA sequencing (Macrogen, Inc., Seoul, Korea).

The DNA plasmid constructs were calculated for the number of copies as shown below and subsequently prepared for a serial dilution to set up standard curves. Ten-fold dilution series of the plasmid constructs were prepared in sterile distilled water (DW), yielding 8 concentrations ranging from 10^1^ to 10^8^ copies/reaction. The calculation formula for number of copies is as follow [23]:Number of copies/µL = Plasmid concentrations (g/µL) × 6.022 × 10^23^(molecules/mole)            (Length of DNA fragment) bp (× 660) Daltons

### 2.3. Optimization of Real Time Quantitative PCR (qRT–PCR)

Optimal conditions of conventional PCR to amplify fragments of *KIR2DL4*, *3DL3* and *RPII* were performed with the cDNA of NK92 using an Veriti^TM^ 96-well fast thermal cycler (Applied Biosystems, Waltham, MA, USA). Thermal cycles were set as follows: one cycle of 5 min at 95 °C, 30 cycles of 30 s at 95 °C, 30 s at 58 °C and 30 s at 72 °C, followed by a 5 min final extension at 72 °C.

Real-time PCR, using the AccuPower^®^ Plus DualStarTM qPCR PreMix MasterMix (Bioneer, Daejeon, South Korea), was optimized using different annealing temperatures based on the end-point PCR and performed by using the Exicycler^TM^ 96 machine. Each 50-µL reaction contained 25 µL 2× AccuPower^®^ Plus DualStarTM qPCR PreMix MasterMix, primers and probes at a concentration of 0.4 µM, 1× of 50× ROX dye and 5 µL of 10^8^ copies/µL of plasmid–*KIR2DL4*, -*KIR3DL3* and -*RPII*. The cycling conditions were an initial first cycle of 5 min at 95 °C, followed by 45 cycles consisting of first 20 seconds at 95 °C and then 45 s at 56.11, 57.78 and 60 °C. An auto-base lining default was used to show the *C*_t_ values. Multiplex PCR reactions were also carried out and the PCR amplification conditions were the same as that of the singleplex assay (annealing temperature at 58 °C).

### 2.4. Sensitivity and Reproducibility

Singleplex real-time PCR assays were carried out prior to multiplexing for all the three primer pairs to evaluate their sensitivity and specificity. For *KIR2DL4*, *3DL3* and *RPII*, serial ten-fold dilutions of plasmid constructs 10^1^ to 10^8^ copies/µL were tested according to the previous protocol. The cycling conditions were 95 °C for 5 min, followed by 95 °C for 20 s and 58 °C for 45 s, for 45 cycles and then, run in triplicates of 6 times for the between run assays. The different plasmid concentrations (copies) were plotted against the corresponding *C*_t_ values. The method efficiencies were calculated according to E = (10^(−^^1^^/^^Slope^^)^ − 1) × 100%.

An irrelevant DNA sequence cloned into the pGemT plasmid in the concentration at 10^8^ copies/µL was simultaneously tested in duplicate with at least six independent experiments for *KIR2DL4*, *KIR3DL3* and *RPII*. Mean *C*_t_ value and standard deviation were calculated and the limit of detection (LOD) was determined, accordingly. Sensitivity of the multiplex was obtained by using the serial dilutions of three plasmid constructs from *KIR2DL4*, *KIR3DL3* and *RPII*. Mixed three templates of 10^2^ to 10^6^ copies of *KIR2DL4* and *KIR3DL3* were simultaneously evaluated along with the baseline levels of *RPII* at 10^2^ and 10^3^ copies in triplicates of 6 times. The mean *C*_t_ value for each dilution was compared between the singleplex and multiplex assays.

### 2.5. Specificity

In cross-reaction assessment, the set of primers and probe of *KIR2DL4* was tested with *KIR3DL1*, *KIR3DL3* and *RPII* plasmids while the set of *KIR3DL3* was tested with the *KIR3DL1*, *KIR2DL4* and *RPII* plasmids in the concentrations at 5 × 10^6^ to 5 × 10^8^ copies per reaction. Moreover, primers and probes of *RPII* were tested with the *KIR2DL4*, *KIR3DL1* and *KIR3DL3* plasmids, accordingly. The data were obtained from duplicates and representatives of three independent experiments.

### 2.6. Human Sample Collection and RNA Extraction

One hundred whole blood samples of healthy donors were received from the blood bank center at Srinagarind Hospital of Khon Kaen University under ethical approval of the Khon Kaen University, Ethics in Human Research Committee (HE621256). To separate the buffy coat, the whole blood samples were centrifuged at 3500 rpm for 10 min at 4 °C and then discarded the serum and collected only the buffy coat. Total RNA were extracted from each buffy coat of a blood sample by The Pure Link^TM^ RNA Mini Kit (Invitrogen, Carlsbad, CA, USA) according to the manufacturer’s instructions, then 500 ng of RNAs were consecutively conversed to cDNA with the Moloney murine leukemia virus reverse transcriptase (M-MLV RT) kit (Promega, Madison, WI, USA) using an Veriti™ 96-well fast thermal cycler (Applied Biosystems, Waltham, MA, USA).

### 2.7. Sensitivity and Specificity in Human Samples

The human cDNA samples were tested with the singleplex and the multiplex qRT-PCR as described above. Then the expression values of *KIR2DL4*, *KIR3DL3* and *RPII* reference gene were analyzed by ROC analysis, we used the limit of detection (LOD) as a cutoff at 10^2^ copies for the singleplex RT-PCR and 10^3^ copies for the multiplex RT-PCR to select the best sensitivity and specificity.

### 2.8. Statistical Analysis

Quantitative data were presented as mean ± standard deviation (X ± SD). The equation of linear regression and linear correlation coefficient (*R*^2^) was determined with the GraphPad Prism software version 5.0 for Windows (GraphPad Software, San Diego, CA, USA). The receiver-operating characteristic (ROC) curves, specificity, sensitivity and Youden’s index (YI), were analyzed by SPSS version 16 (SPSS, Inc., Chicago, IL, USA). Positive likelihood ratio (+LR), negative likelihood ratio (−LR) and were analyzed by the GraphPad Prism software version 5.0.

## 3. Results

### 3.1. Sequence Analysis and Probe Design

From the IPD database, the *KIR* sequences were highly polymorphic representing several alleles from each locus ranging from 16 to 184 alleles (Appendix A). There were 109 and 166 alleles, respectively, for *KIR2DL4* and *KIR3DL3*. The mRNA sequences of all *KIRs* were compared using the Clustal Omega program. The percentages of sequence identities are shown in Appendix A. They were so similar ranging from 70.69%–96.94%. Thus, it would be difficult to design specific primers that amplify all alleles of each locus and discriminate those of other loci. Consequently, specific probes were designed to improve specificity and the primers were used as described by Tajik el al., 2009 [22]. *KIR2DL4* was closely related to *KIR2DL5A* and shared homology of 88.12%. They were type II KIR2D encoding two extracellular domains with D0 and D2. Likewise, *KIR3DL3* was highly similar to the *KIR3D* genes (87.43% homology to *KIR3DL2*). *KIR2DL4*, which also possesses two immunoglobulin-like domains, shares sequence similarities to *KIR3DL1* and *KIR3DL3* with 75.02% and 72.63%, respectively. Expectedly, *KIR3DL3* was more similar to *KIR3DL1* at 86.70% identity as shown in Appendix A.

Specific probes were designed to hybridize within the region between the primers avoiding complementary with other *KIR* sequences and either primer sequences (Appendix A). Probes should not contain a G nucleotide at their 5′ ends due to an arrangement of quencher reporter fluorescence after cleavage from the 5′-exonuclease activity of the *Taq* enzyme. High specificity and effective probes should be able to detect all alleles of target *KIR* genes and differentiate from other *KIR* genes.

### 3.2. Optimization of Annealing Temperature for qRT-PCR

The amplicon PCR products of targeted *KIR* genes and reference control were amplified from the cDNA of NK92 with specific primers. Amplification products of *KIR2DL4*, *KIR3DL3* and *RPII* were approximately 130, 199 and 143 bp, respectively (Appendix A). Further development to determine an optimal annealing temperature among these primers and probes was conducted at 56.11, 57.78 and 60 °C (see “Materials and methods”). We selected the high efficiency of amplification based upon the threshold cycle time (*C*_t_) and the high intensity of signal for all tests. Thus, the annealing temperature at 58 °C was used which was the same as that used in the end-point PCR (Appendix A).

### 3.3. Assessment of Sensitivity and Reproducibility for Singleplex qRT-PCR

Serial dilutions with varying copy number of standard plasmids generated previously were used to determine the sensitivity and variability of the method. The cycle time (*C*_t_) represented the amount of template present at the beginning of the reaction. The standard curve based on the serial dilutions of *KIR2DL4*, *KIR3DL3* and *RPII* clones showed a linear relationship between log copy number and mean of the threshold cycle time (*C*_t_) (Figure 1 and Appendix A). The results obtained from qRT-PCR were significantly high precision or less variable. From the between run experiments, percentages of coefficient of variation (% CV) of all assays were less than 7% even when % CV values of within run and between run should ideally be less than 10% and 15%, respectively. Mean of cycle time (*C*_t_), standard deviation (SD) and percentage of coefficient of variation (% CV) were calculated for each level and were given in Appendix A. However, the reproducibility at low copy number was not so reliable. To determine the limit of detection (LOD), irrelevant sequences were evaluated. The threshold cycle times of irrelevant sequences were overlapping with the standard target template at 50 copies for *KIR3DL3* and at less than 50 copies for the *KIR2DL4* and *RPII* assays (Figure 1A,C,E). Thus, the LOD of *KIR3DL3* was at 500 copies, but LOD of *KIR2DL4* and *RPII* were at 50 copies. The standard curves were generated to show the sensitivity of these assays with the linear regression equation, a square regression coefficient (*R*^2^) and amplification efficiency (E) (Figure 1B,D,F). In theory, correlation coefficient (*R*^2^) for a detection assay should be >0.98 and efficiencies (E) range from 90% to 110%. The linearity of standard curves of *KIR2DL4* and *RPII* presented the lower detection limit at 500 copies because detection assay became less reliable at a lower template concentration of 50 copies per reaction. Therefore, the qRT-PCR assays for *KIR2DL4*, *KIR3DL3* and the reference gene *RPII* had the limit of detection (LOD) at 500 copies. Comparing to the conventional end-point RT-PCR, the detections of *KIR2DL4* and *KIR3DL3* by the developed method were 10 times more sensitive. The conventional method had the detection limit at 10^3^ copies (Appendix A). In addition, the qRT-PCR detection of *RPII* was 1000 times more sensitive than the conventional method with a lower detection limit at 10^5^ copies (Appendix A).

### 3.4. Assessment of Specificity for Singleplex qRT-PCR

To evaluate the specificity, a combination of both specific primers and/probe was tested against the reference *RPII* gene, the irrelevant sequence clones and the related non-target *KIR* sequences. In the cross-reaction experiments, the *KIR2DL4* and *KIR3DL3* detection assays could positively cross-react with the related-KIR genes. This was noted only when the non-target *KIRs* were present at a level of more than or equal to 5 × 10^7^ copies (Figure 1A,C and Appendix A). This very high level of cross-reactivity is insignificant because normally the *KIR* expressions would be less than 10^5^ copies [24]. In contrast, the *RPII* detection assay was specific and did not cross-react with *KIR2DL4*, *KIR3DL1* and *KIR3DL3* (Figure 1E and Appendix A).

### 3.5. Development of Multiplex qRT-PCR

The performance of multiplex qRT-PCR was compared to the singleplex assay. Following limit of detection and specificity of the singleplex, the analytical sensitivity of multiplex real-time PCR was determined on mixtures of serial dilutions between the lower and upper limit of detections (10^2^–10^6^ copies) for *KIR2DL4* and *KIR3DL3* with 10^2^ and 10^3^ copies of the reference gene. The assays were evaluated in triplicate using multiplex conditions as outlined in the Materials and Methods. A competitive amplification was found when 10^5^ and 10^6^ template copies of *KIR2DL4/3DL3* were determined with the *RPII* gene at 10^2^ copies, resulting in no signal of *RPII*. The same results were obtained, although the *RPII* primer and probe concentrations in the reactions were increased to 0.4 and 0.8 µM (Appendix A). A limiting assay was performed to determine at which 10^3^ copies of *RPII* in the multiplex was efficiently amplified with a correlated *C*_t_ in the singleplex (Appendix A). In accordance with this evaluation, there was a strong linear regression coefficient of *KIR2DL4* and *KIR3DL3* (*R*^2^ = 0.996 and 0.999, respectively) which was similar to the single step assays (Figure 1B,D). Compared to the singleplex real-time PCR, the multiplex real-time PCR for *KIR2DL4/3DL3* including the reference gene were less sensitive (LOD = 10^3^ copies). A strong correlation between the singleplex and multiplex qRT-PCR detections was found for all: *KIR2DL4* (*R*^2^ = 0.995), *KIR3DL3* (*R*^2^ = 0.996) (Figure 2A,B). The correlations were effectively linearity when *R*^2^ was higher than 0.99.

### 3.6. Detection of KIR Gene Expression in Human Samples

To validate whether the method is applicable to human samples, an ROC analysis was performed on samples of blood donors. One hundred blood samples of healthy donors with ethical approval (HE621256) were used to study. First, each blood sample was extracted for total RNA, next converted into cDNA and then analyzed for *KIR2DL4*, *KIR3DL3* and *RPII* expressions by the multiplex RT-PCR. The median expressions of *KIR2DL4* and *KIR3DL3* were at 1.4 × 10^4^ (40.67–1 × 10^5^) and 1.9 × 10^4^ (51.45–1.7 × 10^6^) copies, respectively. (Appendix A). In ROC analysis, we used the limit of detection (LOD) at 10^2^ copies for the singleplex RT-PCR and 10^3^ copies for the multiplex RT-PCR as a cutoff. The expressions of *KIR2DL4* and *KIR3DL3* as well as ROC curves were presented in Table 2 and Figure 3. Observing the area under the ROC curve of the three genes by multiplex assay, *KIR2DL4* and *KIR3DL3* gave the highest score at 0.998 and followed by *RPII* with the score at 0.984. The sensitivity and specificity of the multiplex qRT-PCR in healthy donors were calculated with 100% and 99.97%, respectively, for *KIR2DL4* and *KIR3DL3*, but the specificity of *RPII* was 99.85%. For the singleplex assay, the area under the ROC curve of *KIR2DL4* gave the highest score at 0.984 and followed by *KIR3DL3* and *RPII* with score 0.968 and 0.952, respectively. The sensitivity and specificity of the singleplex qRT-PCR in healthy donors were calculated with 100% and 99.96%, respectively, for *KIR2DL4* and *KIR3DL3*, but the specificity of *RPII* was 99.88%. The results indicated that the determination of *KIR2DL4*, *KIR3DL3* and *RPII* expressions in 100 healthy donors by using the developed method had high sensitivity and specificity of both multiplex and singleplex RT-PCR assays. These were consistent with the study using the standard target templates.

Further assessment was demonstrated by the best score of sensitivity, specificity and Youden’s index. The positive likelihood ratio (+LR) was used to indicate the probability for a correct analysis of a method. Usually, a number greater than one or a number that near 10 is indicated as an excellently appropriate method. In contrast, the negative likelihood ratio (−LR) was used to indicate the possibility of incorrect analysis of a method. The lower value indicated optimal, usually less than one. The study demonstrated that, for all 3 genes, the *KIR2DL4* and *KIR3DL3* had the best sensitivity, specificity, LRs and YI (Table 3). These results supported the above data which indicated that the developed method had high sensitivity and specificity.

## 4. Discussion

In this study, we successfully set up a quantitative multiplex real-time RT–PCR with fluorescent probes for reproducible detection of the *KIR2DL4* and *KIR3DL3* transcripts. This technique was carefully developed and designed to maximize the sensitivity and specificity. Although the genomic detection of KIRs has been reported [25], few studies had determined *KIR* expressions by a real-time PCR without the verification and characterization of their assays [19,24]. Validation is the critical evaluation of an assay’s performance for the high-homolog KIR gene family. Moreover, normalization was performed to avoid variations in different tissue samples or various cell types by establishment of a stable reference gene, *RPII* [26]. *KIR2DL4* expression has been reported to be expressed in all human NK cells and on all *KIR* haplotypes [18,27]. In contrast, the transcripts of *KIR3DL3* were more specific to the subset of CD56^bright^ NK cells, especially in women’s peripheral blood and decidual NK cells, though, they are the framework genes in all *KIR* haplotypes [19]. The strong sequence conservation of *KIR2DL4* and *KIR3DL3* genes among many primates suggests that these genes serve a function [28]. Previous research has determined allelic variation of *KIR2DL4* affecting the mRNA level and cell surface expression [29]. Using end-point PCR, the *KIR2DL4* transcripts have been reported to be increased by the IL-2 activation [30] and some specific conditions [31]. Moreover, the HLA-G ligand could induce upregulation of the *KIR2DL4* mRNA. In contrast, *KIR3DL3* was poorly investigated for mRNA expressions due to a technical difficulty. IL-2 cytokine stimulations failed to induce *KIR3DL3* expression [32].

Extensive sequence identities of *KIRs* make them so difficult to establish a molecular method specific to a particular *KIR* gene expression [33]. The *KIR3DL3* expressions were detected only after the 2nd round of RT-PCR and some samples were not detectable even after the 2nd round due to the very low level of *KIR3DL3* [19]. It is possible that the *KIR* mRNA levels are so diverse. Thus, a difficulty in amplifying *KIR* transcripts may affect the detection of mRNA in samples resulting in variable data on expressions of *KIR2DL4/3DL3*. Specific primers with specific probes enable specific detections of *KIR2DL4/3DL3* and differentiate from the others increasing the specificity and sensitivity of detections. We have shown that the sensitivities were improved by 10–1000 times.

The accuracy of absolute quantification depends on the accuracy of the standards. The standards of *KIR2DL4/3DL3* and the *RPII* reference gene were shown to be reliable with a high value of square regression coefficient (*R*^2^). For the singleplex, the amplification was effective with the sensitivity defined by the limit of detection at 500 copies per reaction. The cross reactivity was detected in the *KIR3DL3* detection when the *KIR2DL4* and *KIR3DL1* were present at the same level as the study of Brooks CR et al. [24], which showed cross reactivity of *KIR2DL1* against nonspecific *KIR2DL3* at a level of 5 × 10^7^ copies. It was difficult to find the *KIR* transcripts up to the level of 5 × 10^7^ copies per reaction in clinical samples. Whereas the developed multiplex qRT–PCR, was reproducible indicated by the high correlation coefficient (*R*^2^ > 0.99) and showed the limit of detection at 10^3^ copies which should potentially be sensitive enough for biologic detections.

The copy numbers of *KIR2DL4* and *KIR3DL3* mRNA expressions in samples could be calculated from the slope of linear regression of the standard curve. Target *C*_t_ values were directly compared to an internal reference *C*_t_ (*RPII*) and the data were presented as relative expression of the specific mRNA-targets to the internal reference gene. Changes of *KIR* expression levels are significantly related to biologic function on NK cells; therefore, the variable expression of internal reference expression affects the accuracy calculation of quantitative RNA analysis. The amplification efficiencies of the internal reference must be similar to the targeted gene. RNA polymerase II was selected to be an internal reference gene in this study because it has been reported to be consistently expressed in almost all cells and tissues. Moreover, RPII is a crucial enzyme in mRNA transcription and it constantly expresses under the activation experiment [26]. For gene studies, lower-level expression of *RPII* compared to the usual reference gene *GAPDH*, allows the normalization of slightly differences in the amount between the two targeted genes leading to higher sensitivity.

The validated performance of the developed method on human samples by the singleplex and the multiplex RT-PCR showed that both the multiplex and singleplex methods gave the same high value of sensitivity, specificity and AUC. Moreover, the results of these methods showed that both *KIR2DL4* and *KIR3DL3* were so diversely expressed and the highest expression were 1 × 10^5^ and 1.7 × 10^6^ copies, respectively. Furthermore, *KIR3DL3* mRNA level was higher in female than in male samples but were not statistically significant (median of male = 3.4 × 10^4^ range 51.45–1.4 × 10^5^ copies, median of female = 3.67 × 10^4^ range 50.21–1.7 × 10^6^ copies). This was similar to the study of Trundley AE et al. [19]. Additionally, *KIR2DL4* and *KIR3DL3* were higher in donor samples who were older than 34 years but were not statistically significant. To confirm this result, a larger sample size would be required. The ROC analysis of these samples indicated that the method could be applicable on human samples and was consistent with the standard target templates tested.

## 5. Conclusions

In conclusion, we developed quantitative and specific detections of *KIR2DL4* and *KIR3DL3* expressions based on a multiplex qRT-PCR with fluorescent probe assay. The method was validated using human samples, establishing that the method is applicable to human samples. *KIR* transcriptional levels related to surface protein expressions could control the definite NK functions. This method may aid in determining the physiological and pathologic conditions that *KIR2DL4* and *KIR3DL3* could be expressed on NK cells and subset of NK cells especially in the decidual NK cells.

## Figures and Tables

**Figure 1 diagnostics-10-00588-f001:**
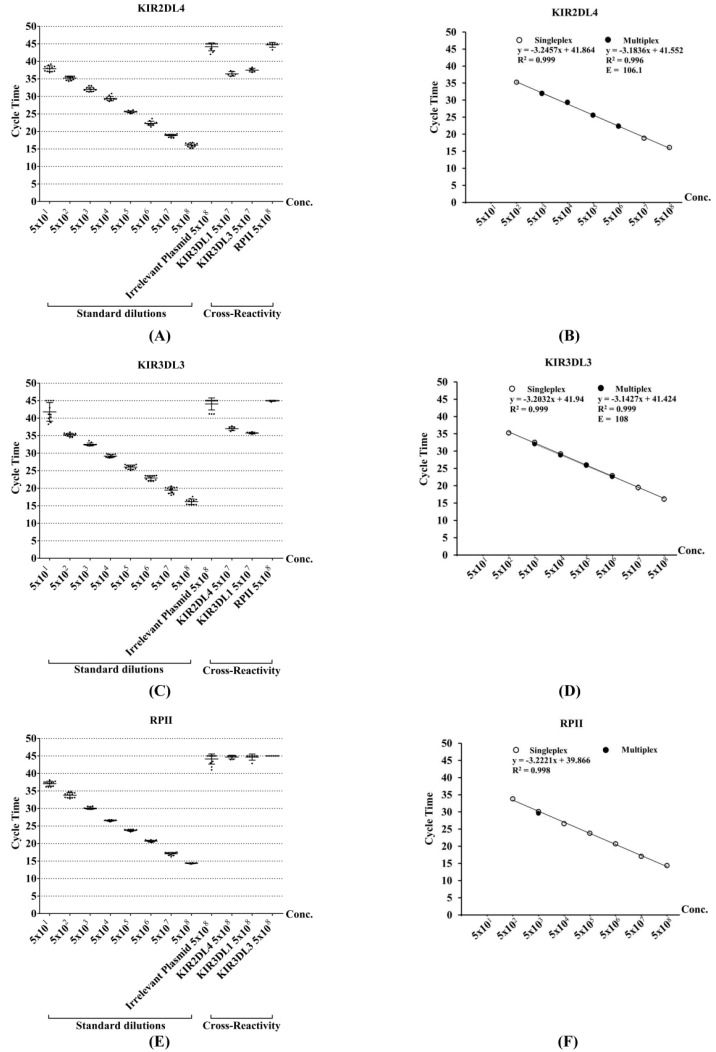
Characterization of the quantitative real-time reverse transcription polymerase chain reaction (qRT-PCR) with fluorescent probe assays to detect *KIR2DL4*, *KIR3DL3* and *RPII*. Left panel shows the scatterplot of the mean cycle time (*C*_t_) for each concentration of the targets, an irrelevant sequence plasmid, *RPII* and non-target *KIR* plasmids to evaluate the sensitivity, specificity and reproducibility of primers/probes for (**A**) *KIR2DL4* (**C**) *KIR3DL3* and (**E**) *RPII* detections; The right panel shows the plot of *C*_t_ against the log of the copy number standards. The multiplex reactions showed comparable efficiencies as the singleplex reactions for (**B**) *KIR2DL4* (**D**) *KIR3DL3* and (**F**) *RPII*. The standard curves are shown with the equation of linear regression, a square regression coefficient (*R*^2^) and the efficiency (E) for qRT-PCR.

**Figure 2 diagnostics-10-00588-f002:**
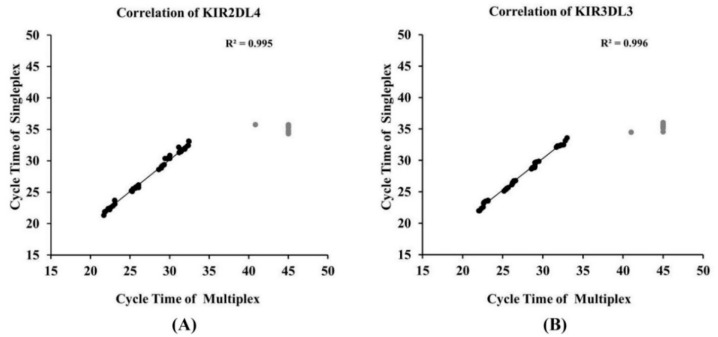
Correlation between the *C*_t_ values obtained with singleplex and multiplex qRT-PCR assays. The coefficient of correlation (*R*^2^) values for *KIR2DL4/3DL3* detections, excluding the outliers marked in gray, were 0.995 and 0.996 for (**A**) *KIR2DL4* and (**B**) *KIR3DL3*, respectively. Higher variations of the *C*_t_ values were found if gene templates were fewer than 500 copies for the multiplex, the outlier data which was out of limit of detection (LOD).

**Figure 3 diagnostics-10-00588-f003:**
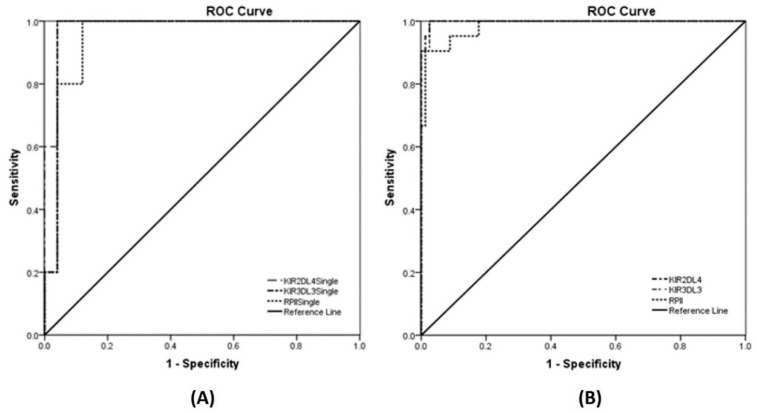
ROC analyses of KIR gene and reference gene expressions in 100 healthy donors were shown. (**A**) ROC curve of the singleplex qRT-PCR showed AUC value for *KIR2DL4* and *KIR3DL3* were 0.984 and 0.968, respectively (*p* < 0.001) and for *RPII* reference was 0.952 (*p* < 0.002); (**B**) ROC curve of the multiplex qRT-PCR, AUC value for *KIR2DL4* and *KIR3DL3* were 0.998 (*p* < 0.001) and for *RPII* reference was 0.984 (*p* < 0.001). ROC—receiver-operating characteristic; AUC—area under the ROC curve.

**Table 1 diagnostics-10-00588-t001:** Primer and probe sequences used in this study.

Probe	5′ Fluorophore–Sequence–Quencher 3′	Position (Exon)	Product Size (bp)	Reference
Specific probes for quantitative real–time RT–PCR				
P-2DL4	FAM–TGGGTTTAACATCTTCACGCTGTAC–BHQ1	89–113 (E3)		This study
P-3DL3	TexasRed–TGCGGGTTCCCAGGTCAACTATTCCATGG–BHQ2	164–193 (E4)		This study
P-RPII	TET–TGCTGGACCCACCGGCATGTTCTTTGG–BHQ1	180–187 (E26),1–19 (E27)		This study
**Primer**	**Primer sequences 5′–3′**			
Specific primers for quantitative real-time RT–PCR				
2DL4	F: TCAGGACAAGCCCTTCTGC	5–23 (E3)	130	[20]
	R: GACAGGGACCCCATCTTTC	116–134 (E3)		
3DL3	F: GCAATGTTGGTCAGATGTCAG	71–91 (E4)	199	[20]
	R: AGCCGACAACTCATAGGGTA	250–269 (E4)		
RPII	F: AGTATGGCATGGAGATCCCC	137–156 (E26)	143	This study
	R: ATAGGCAGGGGTTGCACC	75–92 (E27)		

**Table 2 diagnostics-10-00588-t002:** Area under the ROC curve analysis of killer-cell immunoglobulin-like receptors (KIR) gene expressions in healthy donors.

Genes	AUC (95% CI)	SE	*p-*Value
Singleplex qRT–PCR			
KIR2DL4	0.984 (0.945, 1.000)	0.020	0.001
KIR3DL3	0.968 (0.903, 1.000)	0.033	0.001
RPII (reference gene)	0.952 (0.876, 1.000)	0.039	0.002
Multiplex qRT–PCR			
KIR2DL4	0.998 (0.994, 1.000)	0.002	<0.001
KIR3DL3	0.998 (0.992, 1.000)	0.003	<0.001
RPII (reference gene)	0.984 (0.964, 1.000)	0.011	<0.001

AUC—area under the ROC curve; SE—standard error.

**Table 3 diagnostics-10-00588-t003:** Assessment index of the KIR gene expressions in healthy donors based on ROC analysis results.

Genes	Sensitivity (%)	Specificity (%)	+LR	−LR	YI
Singleplex qRT-PCR					
KIR2DL4	100.000	99.960	25	0.040	99.960
KIR3DL3	100.000	99.960	25	0.040	99.960
RPII (reference gene)	100.000	99.880	8.33	0.120	99.880
Multiplex qRT-PCR					
KIR2DL4	100.000	99.975	40	0.025	99.975
KIR3DL3	100.000	99.975	40	0.025	99.975
RPII (reference gene)	100.000	99.855	6.896	0.145	99.855

+LR—positive likelihood ratio; −LR—negative likelihood ratio; YI—Youden’s index.

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
