# Peer review of "Quantitative Multiplex Real-Time Reverse Transcriptase–Polymerase Chain Reaction with Fluorescent Probe Detection of Killer Immunoglobulin-Like Receptors, KIR2DL4/3DL3"

_diagnostics, 2020, doi:10.3390/diagnostics10080588_

Round 1

Reviewer 1 Report

The authors proposed an update of the method for amplification of KIR2DL4/KIR3DL3 transcripts. I hope the revision of following points could help to improve the manuscripts.

MAJOR COMMENTS:

  1. The author should describe improvement and strong point of the newly proposed method more clearly and concretely. I understood the novel method has similar reliability with previous one, but wonder what was improved and how the detection of the KIR2DL4/KIR3DL3 expression will be changed by the proposed method. For example, if the major improvement was sensitivity, please clearly show how improved sensitivity of the novel method than the previous one with appropriate statistical test (ex. t-test, x2-test)
  2. In Table 4, Table 5, and Figure 3, the multiplex and singleplex method should be compare in sensitivity, specificity, and AUC.

MINOR COMMENTS:

  1. Table 2 and Table 3 could be a supplemental table. 

Author Response

Point 1: The author should describe improvement and strong point of the newly proposed method more clearly and concretely.I understood the novel method has similar reliability with previous one, but wonder what was improved and how the detection of the KIR2DL4/KIR3DL3 expression will be changed by the proposed method. For example, if the major improvement was sensitivity, please clearly show how improved sensitivity of the novel method than the previous one with appropriate statistical test (ex. t-test, x2-test) 

Response: Thank you for the suggestions. In this study, we focused on the analysis of both transcribed-KIR2DL4 and 3DL3 which might be related to pregnancy conditions. Firstly, the previous report published by Brooks et al. (as ref. 24) has studied individually other KIRs including KIR2DL4 (except KIR3DL3) and NKG2A expressions by a singleplex quantitative PCR. Therefore, we have developed a complete, rapid, and reliable multiplex real-time PCR method for the simultaneous detection of KIR2DL4/KIR3DL3 expressions in one tube. It is very useful with the reducing of sample volume, cost, time and, more importantly, comparable quantitative KIR2DL4/KIR3DL3 expressions of a sample. This is the first report of its kind. Secondly, other publication (as ref. 25) has shown the procedure of high-throughput quantitative genomic PCR to determine the gene copy numbers without method verification of sensitivity, specificity, and the limit of detection. The method was not for expression study. We have developed a new method for expressions with novel probe sequences. In expression study, the designs of binding primers and probe sequences must be in the coding sequences which have more limitation than studying both intronic and exonic sequences. In addition, different combinations of specific primers have different detection limits, so, every mixture should be verified individually. We have carefully reviewed these sections and revised it accordingly as follows:

Page 3, Lines 86-90

After “The purpose of this study was to set up the quantitative real-time RT-PCR with fluorescent probe assay to carry out the KIR2DL4/3DL3 mRNA detections.”, we have added “Novel probes specific for KIR2DL4/3DL3 mRNAs have been developed. The novel method could simultaneously and comparably determine the expressions of KIR2DL4/3DL3 in a single tube of a sample using a new reference gene, RPII. The method has been verified for sensitivity, specificity, limit of detection and cross-reactivity as well as validated on human samples.”

Page 11, Lines 348

“Validation is the critical evaluation of an assay's performance for the high-homologue KIR gene family.” was added.

Page 11, Lines 374-375

“The cross reactivity was detected in the KIR3DL3 detection when the KIR2DL4 and KIR3DL1 were present at the same level as the study of Brooks CR, et al [24], which showed cross reactivity of KIR2DL1 against non-specific KIR2DL3 at a level of 5 x 107 copies.” was added.

Point 2: In Table 4, Table 5, and Figure 3, the multiplex and singleplex method should be compare in sensitivity, specificity, and AUC.

Response 2: We agree with the reviewer and have revised the manuscript accordingly, as follows:

Page 9-11, Lines 324-341

We have re-analyzed and changed now Table 2, Table 3, and Figure 3, (previously table 4 and table 5) by adding the data of singleplex on human samples showing sensitivity, specificity, and AUC of both singleplex and multiplex methods as suggested.

Page 9, Lines 301-311

We added “(102 copies for the singleplex RT-PCR and 103 copies for the multiplex RT-PCR)” after “In ROC analysis, we used the limit of detection (LOD)”

We have modified “Observing the area under the ROC curve…” to “Observing the area under the ROC curve of the three genes by multiplex assay, KIR2DL4 and KIR3DL3 gave the highest score at 0.998 and followed by RPII with the score at 0.984. The sensitivity and specificity of the multiplex qRT-PCR in healthy donors were calculated with 100% and 99.97%, respectively, for KIR2DL4 and KIR3DL3 but the specificity of RPII was 99.85%. For the singleplex assay, the area under the ROC curve of KIR2DL4 gave the highest score at 0.984 and followed by KIR3DL3 and RPII with score 0.968 and 0.952, respectively. The sensitivity and specificity of the singleplex qRT-PCR in healthy donors were calculated with 100% and 99.96%, respectively, for KIR2DL4 and KIR3DL3 but the specificity of RPII was 99.88%. The results indicated that the determination of KIR2DL4, KIR3DL3 and RPII expressions in 100 healthy donors by using the developed method had high sensivity and specificity of both multiplex and singleplex RT-PCR assays.”

Page 12, Lines 391-395

We have modified “The validated performance…..” to “The validated performance of the developed method on human samples by the singleplex and the multiplex RT-PCR showed that both the multiplex and singleplex methods gave the same high value of sensitivity, specificity and AUC. Moreover, the results of these methods showed that both KIR2DL4 and KIR3DL3 were so diversely expressed and the highest expression were 1x105 and 1.7x106 coppies, respectively.”

Point 3: Table 2 and Table 3 could be a supplemental table.

Response 3: As advised, we have changed Table 2 and Table 3 as Table S2 and Table S3, respectively, in a supplementary file. Thus, the manuscript has been modified accordingly, as follows:

Page 5, Lines 196

“table 2” was changed to “table S2” in the sentence ”The percentages of sequence identities were shown in table 2.” to read “”The percentages of sequence identities were shown in table S2.”

Page 6, Lines 204-205

“table 2” was changed to “table S2” in the sentence “Expectedly, KIR3DL3 was more similar to KIR3DL1 at 86.70% identity as shown in table 2.” to read “Expectedly, KIR3DL3 was more similar to KIR3DL1 at 86.70% identity as shown in table S2.”

Supplementary materials: Page 2-3, Table S2 and S3 were added. Table S3 was changed to “Table S4”

Page 6, Lines 227, 232

“Table 3” was changed to “Table S3”

Page 8, Lines 265, 268

“Table 3” was changed to “Table S3”

Page 8, Lines 279

“Table S3” was changed to “Table S4”

Reviewer 2 Report

Interesting approach for assessing KIR2DL4/DL3; however, some issues have been raised in the manuscript. The first issue refers to the basal amount of both genes in the cell line under normal conditions and in the presence of IL-2 since the cytokine is required for cell culture. The second issue is concerned based on the fact that based on the template copies of table 3 is difficult to believe that the specificty and selectivity is above 90 % specilly starting with 1000 or less copy. It is also difficult to believe the correlation about singleplex and multiplex. Based on the supplementary figure is also difficult to believe the specificty of the donors to be 100 %. The reason is that you have a mixed type of circulalting NK cells, then, the cells were not purified so the contribution of T cell could heve been higher. There is a contradiction, DL4 is less expressed than DL3 in the human samples so why not analyze the importance of DL3? How do the authors explain the possibility of subpopulations with different expression of DL3 and DL4? Finally, the abstract should state that the method was standarized with the cell line. 

Author Response

Point 1: Interesting approach for assessing KIR2DL4/DL3; however, some issues have been raised in the manuscript. The first issue refers to the basal amount of both genes in the cell line under normal conditions and in the presence of IL-2 since the cytokine is required for cell culture.

Response 1: Interleukin-2 participates in promoting NK cell maturation, activating cytokine production. Indeed, it has not been fully clarified in inducing KIR expression by IL-2. KIR2DL4 is the only member of the KIR family reportedly expressed in every NK cell clone and can be up-regulated by IL-2. Importantly, the same activation potential was not observed for KIR3DL3 in NK92 or decidual NK cells. Therefore, KIR2DL4 and 3DL3 show different expression level. Anyway, our manuscript has no intention in studying the mechanism or effect of cytokines on KIR expression. We only presented the development of method for detections of KIR2DL4/DL3 expressions using an NK cell line RNA (NK92) for setting up.

Point 2: The second issue is concerned based on the fact that based on the template copies of table 3 is difficult to believe that the specificity and selectivity is above 90 % specifically starting with 1000 or less copy. It is also difficult to believe the correlation about singleplex and multiplex.

Response 2: As shown on Page 6, 3.3 Assessment of sensitivity…, the limit of detection (LOD) for all genes, KIR2DL4, KIR3DL3 and RPII are 500 copies for the singleplex method and 1,000 for the multiplex method. Additional data of comparison of the detection of the singleplex and multiplex assays have now been added in Table 2, Table 3 and Figure 3. Please see additional explanation from response to point 2 of the reviewer 1. To clarify the issue, we have revised our manuscript accordingly as responses to reviewer 1 above.

Point 3: Based on the supplementary figure is also difficult to believe the specificty of the donors to be 100 %. The reason is that you have a mixed type of circulalting NK cells, then, the cells were not purified so the contribution of T cell could heve been higher.

Response 3: We have developed the specific primers and probes to KIR2DL4, KIR3DL3 and RPII. It does not mean to evaluate KIR expression in only NK cells but can analyze KIR2DL4/3DL3 in both NK and T cells or other cells in the samples depending on sample preparation i.e. PBMC, tissues, etc. If an investigator would like to determine a specific cell population, that specific cell population would have to be firstly isolated before employment of our method to quantify the expressions of KIR2DL4/KIR3DL3.

Point 4: There is a contradiction, DL4 is less expressed than DL3 in the human samples so why not analyze the importance of DL3?

Response 4: Thank you for suggestions. It is an interesting data leading the new angle for further study. Although, mRNA of KIR3DL3 could be detected but not protein expression. We have published a paper on miRNA regulation of KIR3DL3 (Nutalai R, Gaudieri S, Jumnainsong A and Leelayuwat C. Regulation of KIR3DL3 Expression via miRNA. Genes 2019, 10(8), 603; https://doi.org/10.3390/genes10080603).

Point 5: How do the authors explain the possibility of subpopulations with different expression of DL3 and DL4?

Response 5:  It is still unclear how different KIR repertoires are generated from the same genetic template but the distribution of KIRs appears to be partly determined by DNA methylation releasing a broad range of functional responses. Moreover, in our previous research we have published a novel mechanism controlling KIR3DL3 expression by miRNAs in “Genes” (2019).

Point 6: Finally, the abstract should state that the method was standarized with the cell line.

Response 6: “using NK92” was added to line 29 of the abstract. We have used the standard plasmid constructing from cell line to validate the developed multiplex real-time PCR method. For standardization, RPII was used to standardized as a reference in this assay. Additional validation on human samples have also been presented in the manuscript.

Additional information for point 2 responses of reviewer 2 (the same info provided for review 1)

Reviewer 1

Point 2: In Table 4, Table 5, and Figure 3, the multiplex and singleplex method should be compare in sensitivity, specificity, and AUC.

Response 2: We agree with the reviewer and have revised the manuscript accordingly, as follows:

Page 9-11, Lines 324-341

We have re-analyzed and changed now Table 2, Table 3, and Figure 3, (previously table 4 and table 5) by adding the data of singleplex on human samples showing sensitivity, specificity, and AUC of both singleplex and multiplex methods as suggested.

Page 9, Lines 301-311

We added “(102 copies for the singleplex RT-PCR and 103 copies for the multiplex RT-PCR)” after “In ROC analysis, we used the limit of detection (LOD)”

We have modified “Observing the area under the ROC curve…” to “Observing the area under the ROC curve of the three genes by multiplex assay, KIR2DL4 and KIR3DL3 gave the highest score at 0.998 and followed by RPII with the score at 0.984. The sensitivity and specificity of the multiplex qRT-PCR in healthy donors were calculated with 100% and 99.97%, respectively, for KIR2DL4 and KIR3DL3 but the specificity of RPII was 99.85%. For the singleplex assay, the area under the ROC curve of KIR2DL4 gave the highest score at 0.984 and followed by KIR3DL3 and RPII with score 0.968 and 0.952, respectively. The sensitivity and specificity of the singleplex qRT-PCR in healthy donors were calculated with 100% and 99.96%, respectively, for KIR2DL4 and KIR3DL3 but the specificity of RPII was 99.88%. The results indicated that the determination of KIR2DL4, KIR3DL3 and RPII expressions in 100 healthy donors by using the developed method had high sensivity and specificity of both multiplex and singleplex RT-PCR assays.”

Page 12, Lines 391-395

We have modified “The validated performance…..” to “The validated performance of the developed method on human samples by the singleplex and the multiplex RT-PCR showed that both the multiplex and singleplex methods gave the same high value of sensitivity, specificity and AUC. Moreover, the results of these methods showed that both KIR2DL4 and KIR3DL3 were so diversely expressed and the highest expression were 1x105 and 1.7x106 coppies, respectively.”

Point 3: Table 2 and Table 3 could be a supplemental table.

Response 3: As advised, we have changed Table 2 and Table 3 as Table S2 and Table S3, respectively, in a supplementary file. Thus, the manuscript has been modified accordingly, as follows:

Page 5, Lines 196

“table 2” was changed to “table S2” in the sentence ”The percentages of sequence identities were shown in table 2.” to read “”The percentages of sequence identities were shown in table S2.”

Page 6, Lines 204-205

“table 2” was changed to “table S2” in the sentence “Expectedly, KIR3DL3 was more similar to KIR3DL1 at 86.70% identity as shown in table 2.” to read “Expectedly, KIR3DL3 was more similar to KIR3DL1 at 86.70% identity as shown in table S2.”

Supplementary materials: Page 2-3, Table S2 and S3 were added. Table S3 was changed to “Table S4”

Page 6, Lines 227, 232

“Table 3” was changed to “Table S3”

Page 8, Lines 265, 268

“Table 3” was changed to “Table S3”

Page 8, Lines 279

“Table S3” was changed to “Table S4”

Round 2

Reviewer 1 Report

The paper thought to became acceptable, although the improvement by the study seem to be slight.

Author Response

Thank you for your comments. There is no further comments to addressed.

Reviewer 2 Report

I think that the manuscript is improved by the changes made. There are some  details in the methodology, specificity and reproducibility, tables S3 and S4 which are important to clear, and they have not been discussed in the response.

Author Response

Response to Reviewer

Point 1: I think that the manuscript is improved by the change made. There are some details in the methodology, specificity and reproducibility, tables S3 and S4 which are important to clear, and they have not been discussed in the response.

Response 1:  Sorry for misunderstanding. For the methodology, specificity and reproducibility, we agree with the reviewer and have revised the manuscript accordingly, as follows:

Page 5, Lines 180-185

We deleted “The samples were used in the multiplex qRT-PCR as described above.”

We added;

2.7 Sensitivity and Specificity in Human samples.

The human cDNA samples were tested with the singleplex and the multiplex qRT-PCR as described above. Then the expression values of KIR2DL4, KIR3DL3 and RPII reference gene were analyzed by ROC analysis, we used the limit of detection (LOD) as a cut-off at 102 copies for the singleplex RT-PCR and 103 copies for the multiplex RT-PCR to select the best sensitivity and specificity.”

Supplementary materials: Page 3, Lines 6

“Table S3.  The cross reactivity and reproducibility assessment.” was changed to “Table S3. The reproducibility and cross reactivity assessment.”.

We have deleted the data of Multiplex qRT-PCR and move into Table S4

Supplementary materials: Page 4, Lines 11

We have added the data of Multiplex qRT-PCR in Table S4

Page 8, Lines 283

We added “(Table S4)” in the sentence “A limiting assay was performed to determine at which 103 copies of RPII in the multiplex was efficiently amplified with a correlated CT in the singleplex (Table S4).”.
